# Age- and diet-associated metabolome remodeling characterizes the aging process driven by damage accumulation

Andrei S Avanesov[1], Siming Ma[1†], Kerry A Pierce[2†], Sun Hee Yim[1], Byung Cheon Lee[1], Clary B Clish[2], Vadim N Gladyshev[1,2]*

[1]Division of Genetics, Department of Medicine, Brigham and Women's Hospital, Harvard Medical School, Boston, United States; [2]Broad Institute, Cambridge, United States

**Abstract** Aging is thought to be associated with increased molecular damage, but representative markers vary across conditions and organisms, making it difficult to assess properties of cumulative damage throughout lifespan. We used nontargeted metabolite profiling to follow age-associated trajectories of >15,000 metabolites in *Drosophila* subjected to control and lifespan-extending diets. We find that aging is associated with increased metabolite diversity and low-abundance molecules, suggesting they include cumulative damage. Remarkably, the number of detected compounds leveled-off in late-life, and this pattern associated with survivorship. Fourteen percent of metabolites showed age-associated changes, which decelerated in late-life and long-lived flies. In contrast, known metabolites changed in abundance similarly to nontargeted metabolites and transcripts, but did not increase in diversity. Targeted profiling also revealed slower metabolism and accumulation of lifespan-limiting molecules. Thus, aging is characterized by gradual metabolome remodeling, and condition- and advanced age-associated deceleration of this remodeling is linked to mortality and molecular damage.

*For correspondence:
vgladyshev@rics.bwh.harvard.edu

†These authors contributed equally to this work

Competing interests: The authors declare that no competing interests exist.

## Introduction

The gradual and irreversible decline in cellular homeostasis that accompanies aging can be delayed by using several methods. For example, lifespan of various model organisms can be extended genetically via perturbation of nutrient sensing, pharmacologically by rapamycin treatment, or nutritionally by dietary restriction (*Bishop and Guarente, 2007*; *Fontana et al., 2010*). Up-regulation of stress response and repair pathways, often linked to conditions of mild stress or hormesis (*Rattan, 2010*), may also promote longevity. On the other hand, the specific molecular processes that accompany the delay in the aging process are not well understood.

It is often discussed that aging involves damage accumulation, therein an increase in DNA mutations, errors in protein synthesis, unwanted posttranslational modifications, metabolite by-products and many other damage forms contribute to the aging process (*Rattan, 2008*). The source of damage is frequently linked to metabolic activities, such as those that produce reactive oxygen species and increase oxidative modifications (*Kirkwood and Austad, 2000*). Damage may also be driven by global metabolic infidelity and engage nearly all biological processes involving synthesis and breakdown of cellular components (*Kirkwood and Austad, 2000*; *Gladyshev, 2013*). The exact mechanisms responsible for tolerance against molecular damage are not fully understood, but studies identified robust transcriptional remodeling that accompanies the aging process (*Lee et al., 1999*; *Zou et al., 2000*; *Pletcher et al., 2002*; *McCarroll et al., 2004*; *de Magalhaes et al., 2009*; *Somel et al., 2010*). How such responses are linked to cumulative damage is not known, since gene expression changes may not

**eLife digest** Signs of aging have been observed in many different species, but the underlying mechanisms are still poorly understood. It is thought that aging is influenced by metabolism. For example, scientists have found that metabolism can lead to the accumulation of byproducts, which may cause damage to cells. Moreover, as organisms get older, the diversity of these byproducts can increase. However, it has proven difficult to measure this cumulative damage.

Avanesov et al. have now tried a different approach and examined the relationship between cellular metabolism, lifespan, and cumulative damage. Male fruit flies were raised on one of two diets—a standard diet, or a restrictive diet that extends their lifespan—and a technique called metabolite profiling was then used to monitor more than 15,000 metabolites in both sets of flies.

Avanesov et al. found that the number of metabolites increased over time, suggesting that damage or mistakes in molecular synthesis increased with age. But the number of metabolites reached a plateau in the oldest flies, even in those whose lifespans were artificially extended. This could be due to cells becoming less active as they get very old.

Avanesov et al. also found that the profile of the metabolites changed in a way that was similar to the way that patterns of gene transcription changed. This suggests that there may be a link between transcription—which is the first step in the process that produces proteins in cells—and metabolism and aging.

reflect immediate biological activities and metabolic fluxes. Another complication in such analyses is that individual damage types (e.g., mutations, lipofuscin accumulation, oxidative modifications) vary across conditions and among species (*Rattan, 2008*; *Edman et al., 2009*; *de Magalhaes, 2012*; *Jonker et al., 2013*). The patterns of damage accumulation are also incompletely defined since many of the previous studies did not analyze samples representing very old ages (*Andziak et al., 2006*; *Edman et al., 2009*; *Jonker et al., 2013*). Yet, insights into the properties of cumulative damage are required for comprehensive assessment of the activity-driven damage models of aging. Addressing these critical questions requires examination of damage composition and heterogeneity and the mortality-associated trends in the accumulation of numerous damage forms under both control and lifespan extending conditions. Aside from damage forms, it is unclear how cellular components generally change as a function of age, whether their patterns mimic the transcriptional responses and whether possible changes in their levels can have causal roles in the aging process.

Metabolite profiling is an emerging method that aims to characterize a large number of small molecules in biological systems and identify proximal markers of biological activity (*Kristal and Shurubor, 2005*; *Patti et al., 2012*; *Kotze et al., 2013*). Recently, metabolite profiling was used to explore metabolic signatures of aging in young vs old mice (*Houtkooper et al., 2011*; *Tomas-Loba et al., 2013*) using a set of markers, thereby confirming that aging is associated with alterations in nutrient sensing, lipid and amino acid metabolism, and redox homeostasis. Another strength of metabolite profiling lies in nontargeted profiling, which enables analysis of thousands of small molecules, albeit of unknown chemical properties. Nontargeted profiling may also be used to characterize patterns of molecular damage by following changes in metabolite diversity. For example, damaged molecular species are expected to exhibit age-associated mass shifts, which may be represented by an increase in metabolite diversity, potentially offering a larger repertoire of damage forms than analyzed previously. By following global changes in metabolome remodeling in response to aging and lifespan-extending interventions, the nontargeted profiling may also lead to the identification of small molecule regulators of the aging process.

Here, we utilized nontargeted, liquid chromatography mass spectrometry (LC-MS)-based metabolite profiling to explore age-associated patterns in metabolite diversity and biological activity in *Drosophila* males maintained on diets that support different lifespan. We discover that metabolite diversity shows a robust age-associated increase. In contrast, known metabolites as well as many metabolites that are detected at all ages show bi-directional trends during aging: they fall predominantly into increasing and decreasing age-associated clusters. Interestingly, older cohorts feature deceleration in both the rise in metabolite diversity and overall cellular activity. Together, these data suggest that changes reflecting biological activity are linked to the display of metabolome diversity,

which is reflected in the appearance of new small molecule species. We followed metabolites implicated in certain forms of damage and found their levels to be higher in long-living flies. These data further implicate a heterogeneous nature of damage whose effect on survivorship is condition dependent. Overall, these metabolite analyses provide critical insights into the activity-driven nature of the aging process.

## Results

### Age-related changes in metabolome diversity follow the lifespan curve

We maintained fruit flies (males) throughout their lifespan on two dietary regimens: standard sugar and yeast diet and fully defined diet (*Mair et al., 2005*; *Lee and Micchelli, 2013*). The defined diet, which was prepared from chemical components and mimicked dietary restriction conditions (*Bass et al., 2007*), led to lifespan extension (*Figure 1A*), similar to the well-characterized effect of dietary restriction (*Mair et al., 2005*). Our design and choice for sampling age groups were also similar to the previous studies, which analyzed gene expression during *Drosophila* aging (*Zou et al., 2000*; *Pletcher et al., 2002*), except that we increased the number of samples at the end of the lifespan curve, including very old flies, which allowed us to better examine changes associated with advanced age.

Our metabolite profiling platform detected >15,000 unique analytes (nontargeted, non-redundant, de-isotoped LC-MS peaks), whose relative abundance was followed as a function of age. We found that the number of detected metabolites was lowest in the young flies in each dietary group and increased as a function of age (repeated measures ANOVA, $p < 6 \times 10^{-6}$ for each diet). On the other hand, the overall signal (the sum of signals for all molecular species detected) slightly decreased throughout lifespan (*Figure 1A*). As metabolites were extracted from the same number of flies at each time point, the decrease in total signal may represent lower biomass in older flies, whereas the number of detected metabolites indicated increased metabolite diversity during the aging process. This pattern was observed in each replicate sample and in each diet.

Interestingly, the rise in the number of detected metabolites was steady during adulthood, yet leveled off (and even slightly decreased) at the most advanced ages (*Figure 1A*). This transition in the metabolite diversity corresponded to the late-life transition in the lifespan curves and mortality (*Figure 1—figure supplement 1*), suggesting relationship between metabolite diversity and the aging process. In contrast, analysis of metabolite diversity using a panel of 205 known metabolites did not show significant age-associated trends (*Figure 1A*). Furthermore, the distribution of signal intensities between targeted metabolites and metabolites, which show diversity fluctuations throughout lifespan were different by few orders of magnitude (*Figure 1B,C*), suggesting that the latter generally corresponded to molecules of low abundance and were not due to errors of detection. Also, changes in metabolite diversity could not be attributable to the presence of yeast-derived metabolites in our diet, since the chemically defined diet contained no cellular products. Lastly, the observed changes in metabolite diversity could be explained by the presence of outliving cohorts in samples near the end of the lifespan curve, as previously proposed for mortality rates (*Curtsinger et al., 1992*). These data suggest a fair degree of synchronization between damage accumulation, expressed by the pattern of metabolite diversity, and the mortality pattern, including their changes in late life.

### Metabolome remodeling defines progression of aging

Among the signals consistently detected in all aging samples (those showing no difference in diversity), there may be metabolites that are passengers or drivers of the aging process, similar to the genes with age-associated differential expression (*Lee et al., 1999*; *de Magalhaes et al., 2009*; *Houtkooper et al., 2011*; *Plank et al., 2012*; *Tomas-Loba et al., 2013*). Using repeated measures ANOVA and stringent statistical cut-off that estimated <0.2% of false positives ('Materials and methods'), we identified 2234 and 2216 metabolites (~14% of all detected metabolites) that qualified as age-associated in the standard and defined diets, respectively. This revealed widespread and dynamic metabolome reorganization throughout lifespan. Of the detected age-associated features, 1,066 were common to both dietary regimens (*Figure 2A*). These molecules exhibited very similar trajectories of change between the two diets (median Pearson correlation coefficient = 0.76), in contrast to the lack of such correlation when using all metabolite pairs between the diets (permutation test p<0.001) (*Figure 2B*). To further characterize these age-related metabolites, we performed two types of scaling: between-group (a metabolite was scaled across both dietary conditions) and within-group (a metabolite was

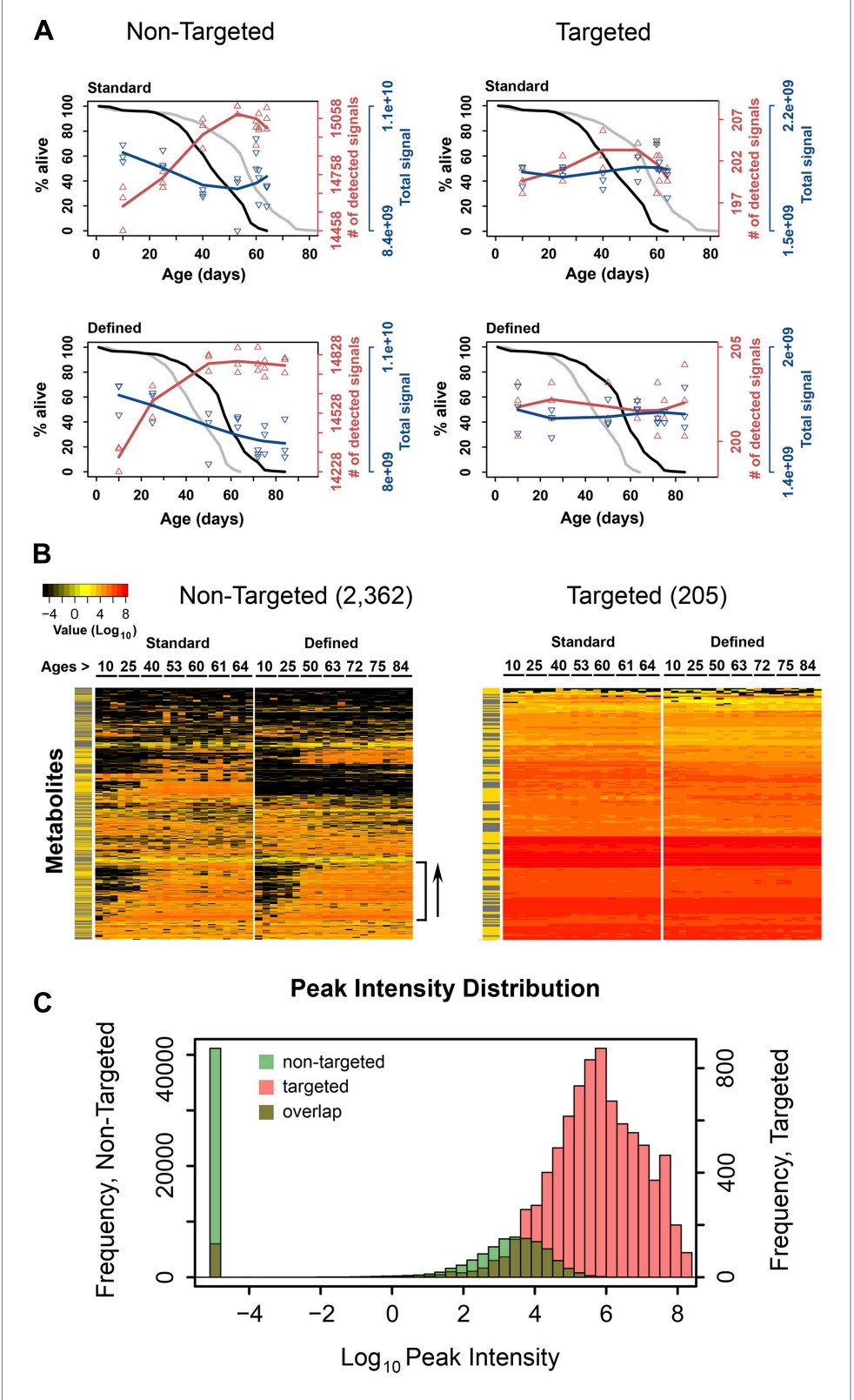

**Figure 1**. Dynamics of metabolite diversity throughout lifespan. (**A**) The number of detected nontargeted metabolites rises and then levels off as a function of cohort's age. Age-dependent changes in the number of detected metabolites (red curve) and intensity of total signal (blue curve) for nontargeted (two left panels) and *Figure 1. Continued on next page*

*Figure 1. Continued*

targeted (two right panels) metabolites for standard (two upper panels) and defined (two lower panels) diets are shown. The lines were drawn using cubic polynomial fit function. Triangles mark data for the separately collected replicates for each age group. Significance for age-associated pattern in metabolite diversity was established using repeated measures ANOVA and was significant for nontargeted metabolites ($p < 6 \times 10^{-6}$) but not significant for targeted metabolites ($p > 0.2$). The corresponding lifespan curves are shown in black in each panel, and the curves in grey (for the other diet) are shown for convenient comparison of survivorship on the two diets. Mean lifespan of flies on standard and defined diets was 50.8 and 64.4 days, respectively (log-rank test [$p < 0.001$]). (**B**) Metabolites which registered at zero in at least one sample (21 total samples [three associated replicates for each of the 7 age groups]) were isolated from the dataset and, for visualization purposes, non-detected signals (ones registering at 0) were changed to $1 \times 10^{-5}$ and $Log_{10}$ transformed with the remaining signals. Accordingly, points of non-detection in black along with the color gradient of the mass-spectrometry peak intensities for detected signals are provided on two age-supervised hierarchically clustered heatmap images. For comparison purposes, only metabolites overlapping in both dietary regimens were used. Side bracket exemplifies rises in age-associated metabolites. Side bars highlight metabolites from the lipid fraction (yellow). Other metabolites are shown in black in this bar. (**C**) Histograms show overlaps in the distribution of signal intensities for all nontargeted metabolites vs targeted metabolites used to construct heat map images in **B**.

The following figure supplements are available for figure 1:

**Figure supplement 1**. Correspondence in late life transition between metabolite diversity and mortality.

scaled within each dietary condition). Principal component analysis (PCA) of the between-group scaled values segregated samples first by diet and then by age (*Figure 2C*), suggesting that the magnitudes of these metabolites were affected by dietary conditions. On the other hand, PCA of the within-group scaled data revealed an age-dependent, but diet-independent segregation, with the replicate

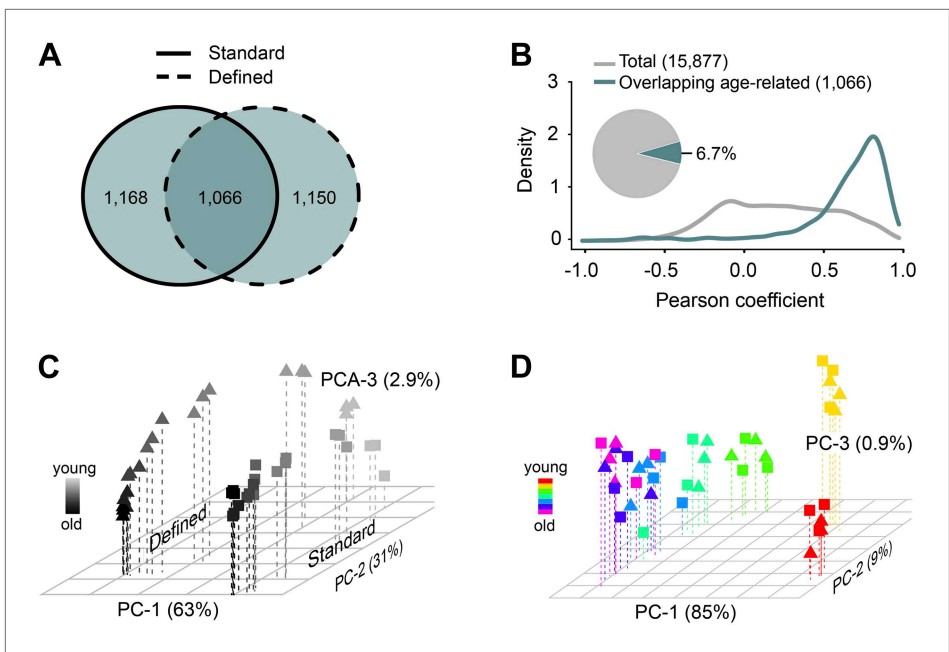

**Figure 2**. Age- and diet-associated changes in metabolite levels accompany *Drosophila* aging. (**A**) Venn diagram of age-related features for two dietary regiments. The diagram shows that a large fraction of detected age-related features overlap between the two diets ('overlapping metabolites'). (**B**) Kernel Density Plot showing the distribution of Pearson correlation coefficients of the overlapping metabolites between the two diets (color). The coefficients for the total metabolites (gray) are shown for comparison. (**C** and **D**) Clustering of the overlapping metabolites by Principal Component Analysis. Squares and triangles denote standard and defined diets, respectively. Each plot includes three replicates per age. (**C**) Plot with scaling expression values across diets. In this plot, individuals are separated according to their ages and diets. (**D**) PCA plot with diet-specific scaling separates individuals according to their ages only.

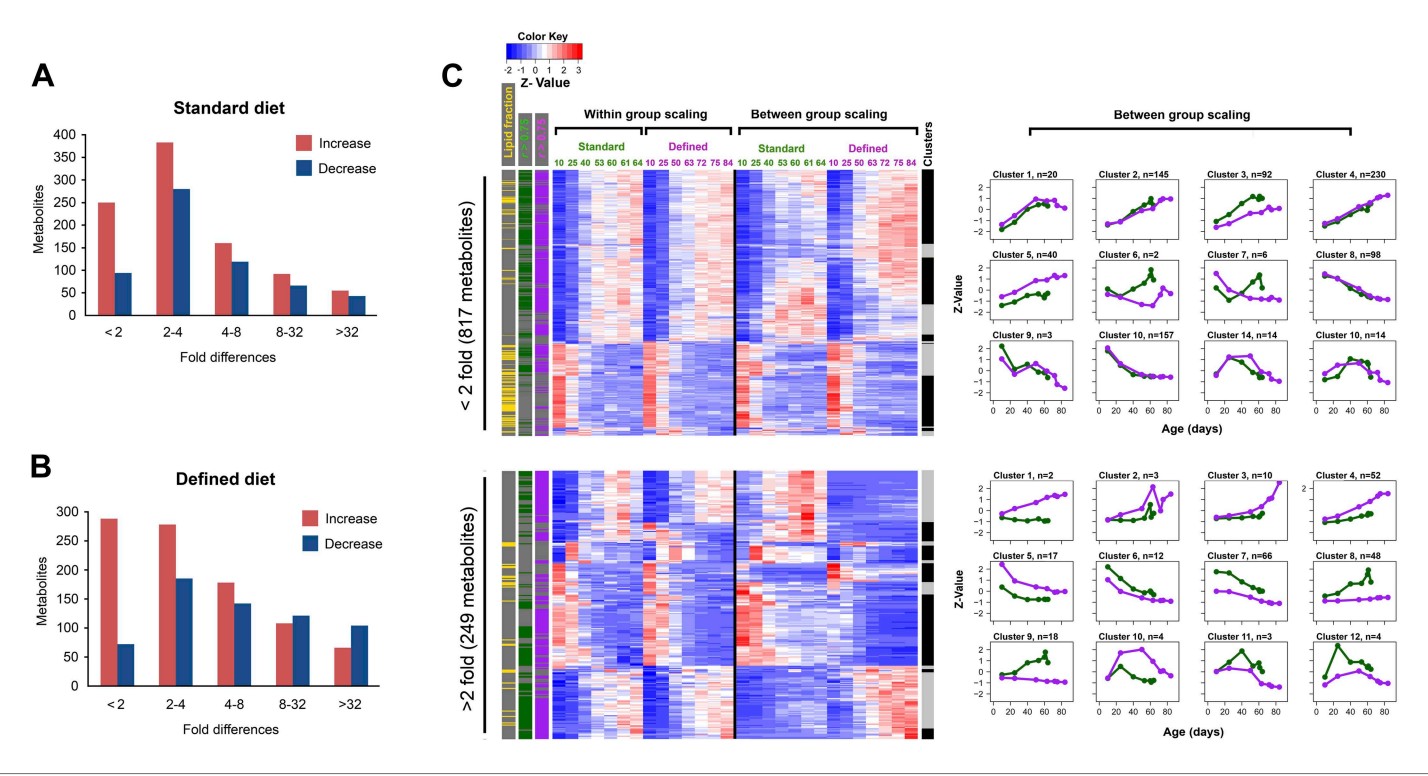

**Figure 3**. Distribution of fold-changes for age-related metabolites. (**A** and **B**) Fold-change differences within standard (**A**) and defined (**B**) diets were calculated by comparing changes in intensity from the ratio of maximum to minimum lifespan-associated values. (**C**) Inter-dietary differences are shown in two heatmap panels after their separation into twofold thresholds, which also show metabolite remodeling during aging. Heatmaps were generated as follows. Replicate values were averaged and then scaled within individual and also across the diets. The resulting matrix was then subjected to age-guided complete hierarchical clustering using hclust algorithm in R where ages were assigned to columns and individual metabolites were assigned to rows. The resulting image allows convenient visualization of clusters containing metabolites with common trajectories (left side), which may also show inter-dietary differences in levels (right side). Side bars were added to highlight metabolites derived from the lipid fraction and also trajectories bearing strong correlation to lifespan curves (Pearson coefficient |r| >0.75, color coded for each diet). Age-related trajectories were derived from trimming the distance matrix into 12 k-means clusters using rect.hclust function in R. Plots in each box represent averages of the scaled values of contributing metabolites whose number is listed in at the top of each graph.

samples of both diets grouping closely together (**Figure 2D**). Thus, these metabolites indeed followed very similar trajectories with age, after normalization of differences in their intensity levels. Furthermore, most age-related changes in metabolite intensity (levels hereafter for simplicity) within individual diets did not exceed 4–8-fold across lifespan (**Figure 3A,B**), which was consistent with the differences in gene expression between *Drosophila* subjected to dietary restriction or stress (**Zou et al., 2000**; **Pletcher et al., 2002**; **Girardot et al., 2004**; **Landis et al., 2012**). These data show that diet- and age-influenced changes in metabolome remodeling throughout lifespan closely recapitulated transcriptional changes and progression of aging.

## Distinct and common age-related metabolite patterns characterize short- and long-lived flies

We explored the relationship between changes in age-related metabolites and longevity. We carried out k-mean clustering to group metabolites with similar trajectories, separately for those with below twofold and above twofold inter-dietary changes (**Figure 3C**), which showed that the diet-influenced changes were relatively underrepresented. Specifically, among the 1,542 age-related metabolites common to both diets, 76% showed less than twofold changes between the two diets (**Figure 4A**), indicating that the influence of diet was noticeable, but mostly small in magnitude. These age-related changes may be linked to longevity as there was high prevalence of strong correlations (both positive and negative) between individual metabolite levels across lifespan and lifespan curves (**Figure 4B**).

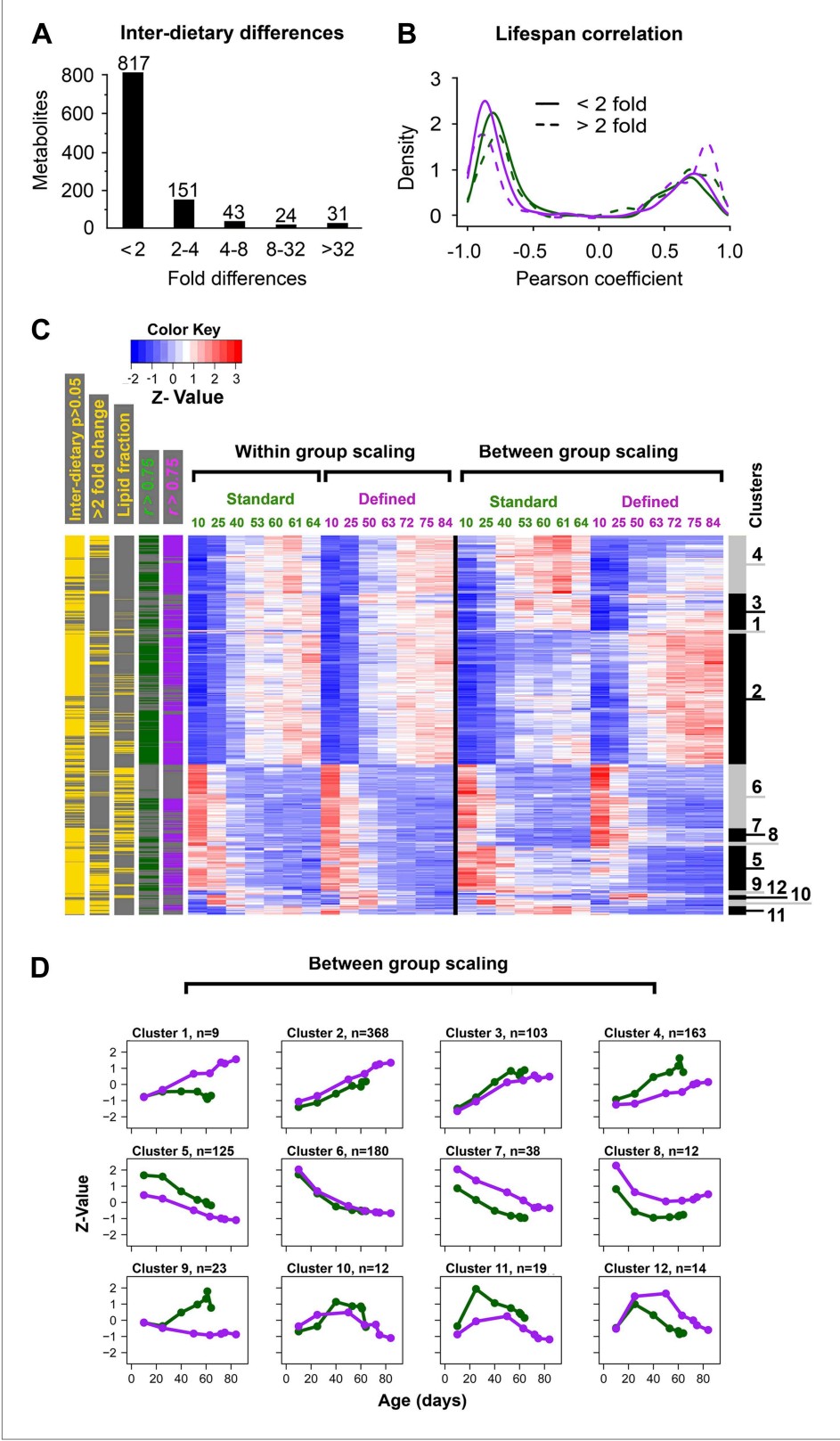

**Figure 4.** Common and distinct patterns of metabolite remodeling during the aging process. (**A**) Fold-change differences between common age-related metabolites from two dietary conditions (inter-dietary differences). Fold change was calculated using averages of individual metabolite levels across each lifespan. (**B**) Distribution of
*Figure 4. Continued on next page*

*Figure 4. Continued*

Pearson coefficients. The Kernel-Density function in R was used to plot the distribution of all Pearson coefficients representing correlations between each of the age-related metabolites and lifespan curves for standard (green) and defined (purple) diets. Signals were split into groups that showed inter-dietary differences of under (solid lines) or above (dashed lines) twofold-change. (**C**) Signals were clustered using methods described in *Figure 3C* legend. Side bars were added to highlight positions of the metabolites bearing statistically significant inter-dietary differences (Student *t* test, p<0.05), metabolites meeting above twofold inter-dietary change, metabolites from the lipid fraction, and metabolites bearing strong correlation to lifespan (Pearson coefficient |*r*| >0.75, color coded for each diet). (**D**) Age-related trajectories were derived from the hierarchical tree as described in *Figure 3C* legend.

While only a small fraction of metabolites showed more than twofold changes between the diets, they typically exhibited strong correlation with lifespan, suggesting they might represent mechanisms through which dietary modification (e.g., dietary restriction) affected longevity. Next, to visualize the changes in metabolite levels during aging, we performed hierarchical clustering on both within-group and between-group scaled data for all signals commonly changing with age in both diets (*Figure 4C*). Interestingly, a very large fraction of metabolites that decreased with age was derived from the lipid fraction, and this observation applied to both diets (*Figure 4C*). Lipid profiles also declined with age in other animals (*Houtkooper et al., 2011*; *Lapierre et al., 2011*; *Singh and Cuervo, 2012*), suggesting that our analysis captures a general scheme of age-related changes in lipid metabolism. Among decreases, metabolites in clusters 6 and 8 declined more rapidly beginning in early, very young samples, suggesting that these signals had an earlier role, for example, during development and morphogenesis. In fact, age-related transcripts with the roles in morphogenesis experienced similar trends (Clusters 6, 7, *Figure 5A,B*). Although most of the decreases in clusters 6 and 8 were significant, they were not strongly different between the diets. This may reinforce the idea of the developmental role for these lipid profiles as flies were reared on the standard medium until day 3 post eclosion, before the transfer to dietary restricted media. In contrast, most polar metabolites increased with age and their rates of change between the diets were more dynamic (*Figure 5C*). In summary, the majority of age-related trajectories were very similar in both diets: they showed strong correlations to lifespan curves and were largely continuous as they followed trajectories observed in younger samples, while very few showed mid-life reversals (clusters 10–12) (*Figure 4D*). Some of the variations in metabolite magnitude between the diets, on the other hand, may be responsible for the observed lifespan differences.

## Similar changes in age-related metabolite and transcript profiles between short- and long-lived flies

If the observed global changes in metabolite levels are biologically relevant, we should expect to see similar patterns in gene expression across lifespan. For this, we analyzed a gene expression data set that sampled mated female flies raised similarly to our two dietary regimens (*Pletcher et al., 2002*). By applying similar statistical methods as those used for identifying the age-associated metabolites, we identified 1172 age-associated genes (ca. 8% of total) common to both diets (repeated measures ANOVA, p<0.0013). Clustering analysis revealed that about 70% of age-related genes also featured gradual transitions throughout lifespan, both increasing (clusters 2 through 5) and decreasing (clusters 6 through 8) with age (*Figure 5A*). We used Gene ontology (GO) (*Ashburner et al., 2000*) to test for enrichment of biological processes within each cluster and found that longevity in dietary restricted (DR) flies was associated with reduced metabolism (cluster 2), increased transport (cluster 11) and altered response to pathogens, albeit early in life (cluster 4) (*Figure 5B*). Similarly to previous studies in mammals and other species, genes with increased levels in cluster 1 were enriched for stress response genes, whose expression rose at slower rate in DR flies, suggesting that improved homeostasis in these individuals may reduce the need for certain stress protection (*Lee et al., 1999*; *de Magalhaes et al., 2009*; *Plank et al., 2012*). Another interesting observation was that all clusters, except for clusters 4 and 5, were enriched for non-overlapping GO terms, suggesting that the age-related changes in distinct biological processes proceeded synchronously. We suggest that this should also apply to nontargeted metabolites.

To compare metabolite and gene expression data further, we examined the frequency of diet-dependent vs diet-independent changes (e.g., trajectories that trended at different rates, levels and

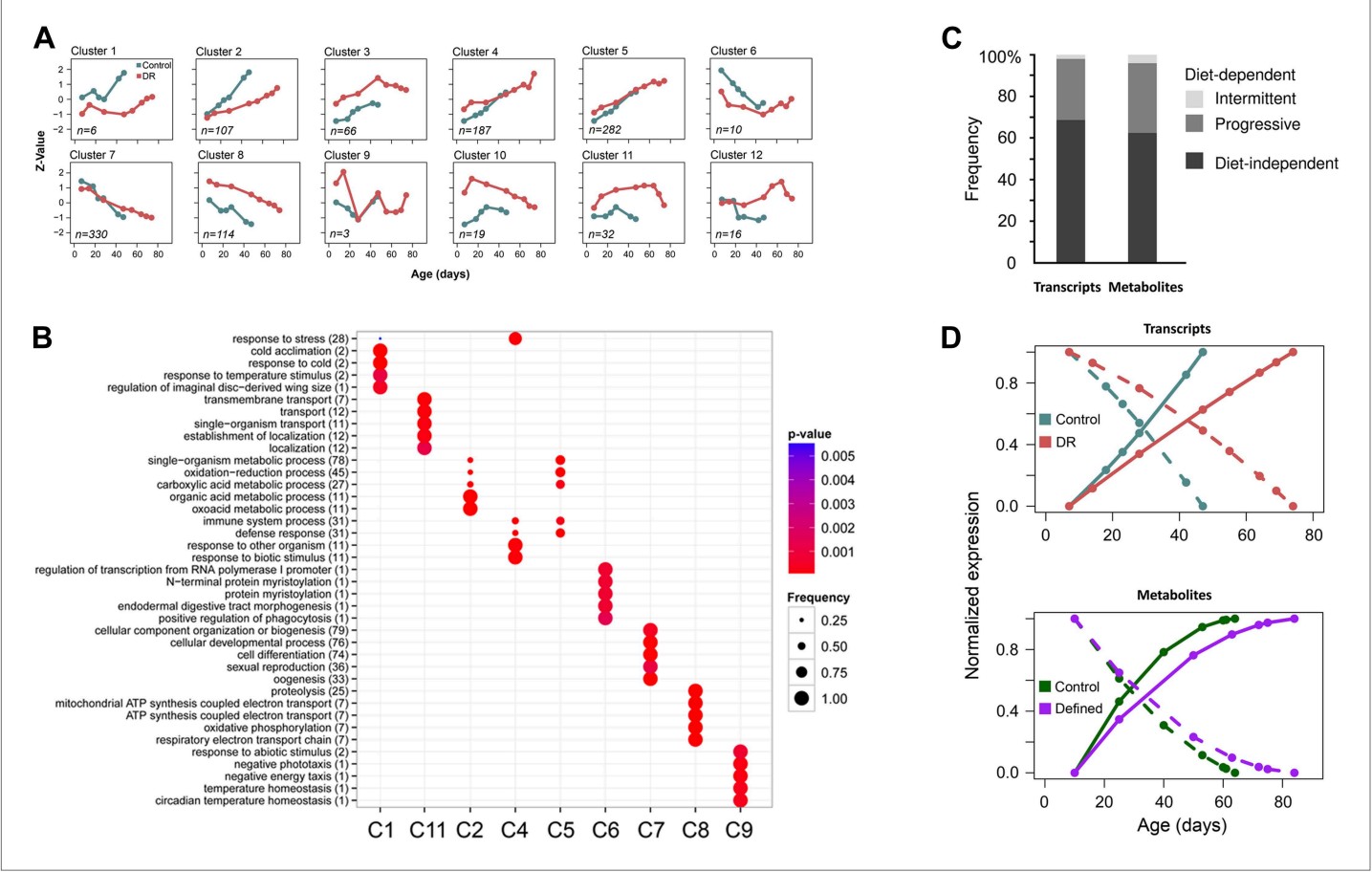

**Figure 5**. Age-related transcripts and metabolites follow similar trajectories and show a delayed response under lifespan-extending dietary conditions. One way repeated measures ANOVA was used to identify transcripts with age-related changes at p<0.0013. A total of 1171 features showed significance in both diets. (**A**) Normalization and clustering were performed according to the procedures described for *Figure 3C* legend. Each box represents individual clusters trimmed from hierarchically clustered tree using hclust algorithm in R. The number of genes contributing to each cluster is provided in the bottom left corner. (**B**) ClusterProfiler (*Yu et al., 2012*) package in R (*Yu et al., 2012*) was used to test for enrichment for Biological Process ontology in Clusters 1–12 in (**A**). Clusters 3, 10, and 12 did not enrich and therefore are not present. (**C**) Comparison of diet-dependent and diet-independent frequencies in gene and metabolite expression data. Frequencies of diet-dependent to diet-independent changes in gene expression and metabolites were obtained from signals provided by clusters 112 in panel **A** and *Figure 4D*, respectively. Differences that were continuous across lifespan were categorized as progressive, and those that were not as intermittent. (**D**) Average trajectories of upregulated (solid lines) and downregulated (dashed lines) signals in gene (top panel) and metabolite (bottom panel) expression datasets. For gene expression, the upregulated trajectories are averages of all signals from Clusters 1–5 shown in panel **A**, while all downregulated signals were derived from Clusters 6–8. Similarly, the global increases and decreases in metabolite levels were generated by averaging signals in Clusters 2–4 and Clusters 5–8, respectively, from *Figure 4D*. Plots show normalized trajectories' values, which were obtained using quadratic polynomial fit through sample replicates. Points indicate sampled ages.

directions between the diets) (*Figure 5C*). First, for transcript changes, there were two types of diet-dependent effects: progressive, which differed at all age points (clusters 1–3, 8, 10 and 11) and intermittent which differed at some, but not all age points (clusters 6, 9 and 12). The diet-independent effects, however, predominated (68%) and fell into clusters 4, 5 and 7. Strikingly, we observed similar frequencies for age-related metabolites whereby clusters 2, 3, 6 and 10 can be categorized as diet-independent making up to 62% of total age-related signals common to both diets. The remaining fraction represented both progressive (clusters 4, 5, 7, 8, and 11) and intermittent (clusters 1, 9 and 12) effects. The diet-dependent differences in metabolite levels were more obvious after the clusters were separated into groups with below and above twofold inter-dietary difference (*Figure 3C*). The observed gradual, dynamic metabolome remodeling throughout lifespan, including both increases and decreases in metabolite signals, implied a role of predetermined, programmatic (but not necessarily programmed), genetically-defined aging (*Pletcher et al., 2002*; *Somel et al.,*

*2010*; *de Magalhaes, 2012*). Importantly, similarities between their trajectories indicate that age-related metabolites effectively recapitulate gradual and continuous aging transitions. We propose that the vast majority of age-related metabolite trajectories are directly related to aging and may contribute to its manifestation and/or delay.

## Coordinated delays in transcriptome and metabolome remodeling in long-lived flies and in late life

We further examined coordination in transcriptome and metabolome remodeling by examining transcript and metabolite signals that changed in both dietary groups. Interestingly, these age-related changes (both increases and decreases common to both diets) in either gene expression or metabolite levels were delayed in long-living flies (*Figure 5D*). Moreover, metabolome remodeling represented by these signals was decelerated with age within each dietary group (i.e., the signals initially showed robust changes, but then almost leveled off). Together with the analyses that revealed the diet-associated changes in the metabolome composition, these findings suggested that a change in lifespan involves both a delay in age-related metabolome and gene expression remodeling and a change in the set of expressed metabolites and genes at any point throughout lifespan. These data also resemble the metabolite diversity pattern (*Figure 1A*) and suggest a close relationship between activity-dependent damage accumulation and aging.

## Lifespan extension is associated with decreased metabolism and tolerance to damage

Behavioral or genetic alterations of nutrient sensing pathways and metabolism lead to lifespan extension in diverse phyla (*Lin et al., 2002*; *Pletcher et al., 2002*; *McElwee et al., 2007*; *Panowski et al., 2007*), suggesting that metabolite profiling could capture metabolic alterations between aging control and long-lived flies. We addressed this by examining several metabolic pathways by screening for age-associated profiles among 205 annotated metabolites, which we recalled from nontargeted spectral features and confirmed using authentic in-house standards. Among these, 81 metabolites displayed age-associated trends under both dietary regimens (repeated one-way ANOVA, p<0.05) and were associated with biosynthesis of amino acids, lipids, energy homeostasis, and damage production (*Figure 6*). As a whole, the clustering analysis by means of inter-dietary comparison of these age-associated molecules showed no prevalence for metabolite downregulation in flies from the dietary restricted regimen: there was nearly an equal frequency of diet-dependent to diet-independent effects (*Figure 7A,B*). Examination of the molecules within individual clusters (as they share similar age-related trajectories) led to several important observations. First, we find that, as in the case of nontargeted metabolite profiles, decreases of targeted signals were predominantly associated with lipids (cluster 1, *Figure 7A,B*). Eighty percent of metabolites in this cluster were triglycerides (TAGs) and diacylglycerols (DAGs), all of which normally rise highly during larval growth but decline thereafter (*Carvalho et al., 2012*). The high developmental expression of these lipid species may explain their rapid drop during adulthood in our data, well before the first major change in survivorship curves. Such patterns are also present within decreasing clusters of nontargeted metabolites that contain a large number of metabolites from the lipid fraction (clusters 6, 8, *Figure 4D*). Also, the age-associated tendencies between our standard and DR-like regimens suggest increased synthesis (or reduced utilization) of lipids in long-living flies and agree with higher lipid stores and starvation resistance of DR flies in previous studies (*Burger et al., 2007*; *Katewa et al., 2012*). In contrast, we observed a progressive age-associated increase in the cholesteryl esters in our long-living flies compared to controls (cluster 11). While accumulation of this lipid class has been inferred to negatively affect mammalian fitness (*Lawton et al., 2008*), it may extend *Drosophila* lifespan via increased pathogen protection (*Caragata et al., 2013*). Second, consistent with the effects of dietary restriction, we find that downregulated metabolites in long-lived flies (clusters 5–8, n = 39) were associated with insulin signaling, energy homeostasis, and amino acid metabolism (*Figure 7C*). Third, a striking observation was the elevation of methionine sulfoxide, which represents damage (*Orentreich et al., 1993*; *Ruan et al., 2002*; *Koc et al., 2004*), and taurine, a product of cysteine degradation and a component of bile acids (*Massie et al., 1989*; *Brosnan and Brosnan, 2006*), in long-lived but not in control flies (cluster 9, *Figure 7D*). The levels of these molecules started to decline at the point corresponding to high mortality in both groups (at 40 and 50 days, respectively), which may be due to the low levels of these metabolites in cohorts surviving to old age (*Curtsinger et al., 1992*). Tryptophan is also a lifespan limiting amino acid

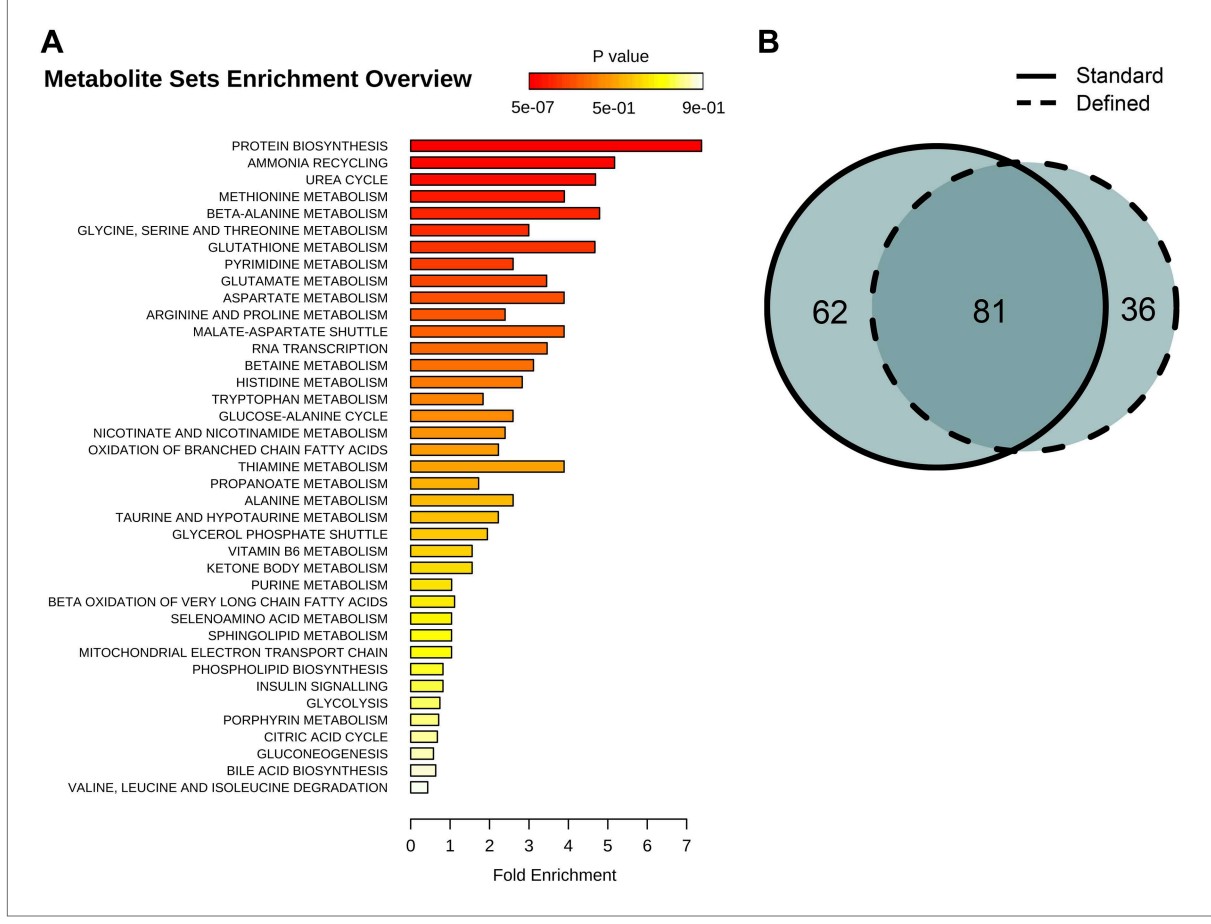

**Figure 6**. Identification and metabolic pathway representation of significant age-related targeted metabolites. (**A**) Overview of molecules significantly associated with aging according to biological processes in both diets (repeated measures ANOVA, p<0.05). (**B**) Venn diagram showing the number of significantly changing metabolites with relation to the number of metabolites uniquely significant to standard (solid) or defined (dashed) diets.

(*De Marte and Enesco, 1986*) and a member of that cluster showing a similar pattern. Such bell-shaped lifespan-associated curves are not surprising as degradation of tryptophan and rise of the downstream metabolite kynurenine (also lifespan limiting metabolite [*Oxenkrug et al., 2011*] expressed higher on the defined diet) were reported in older individuals (*Moroni et al., 1988*; *Frick et al., 2004*). In sum, the higher levels of methionine sulfoxide, tryptophan, and kynurenine in our long-living flies are consistent with the notion that lifespan extension by dietary restricted regimen may increase damage tolerance. We suggest that long-living species may also feature higher levels of lifespan-limiting molecules and that metabolome remodeling upon dietary interventions activates compensatory mechanisms. Whether this is broadly applicable to a variety of lifespan-extending conditions and to long-lived species is a possibility worth exploring.

## Discussion

Aging was suggested to result from accumulation of molecular damage that leads to age-associated decrease in organism's fitness (*Orgel, 1963*; *Kirkwood and Austad, 2000*). Yet, individual damage forms, for example, damage caused by reactive oxygen species, proved to be both condition- and species-specific, challenging the relevance of damage accumulation to aging (*Blagosklonny, 2008*). However, aging may be associated with cumulative damage, whereby mild effects of the myriad damage forms may exhibit additive effects on organism's fitness. In this case, increased mortality may be associated with inevitable accumulation of heterogeneous damage forms, which are also condition dependent. For example, in interventions that extend lifespan (e.g., dietary restriction) reduction of some damage forms may be compensated by accumulation of other forms of damage. In addition to

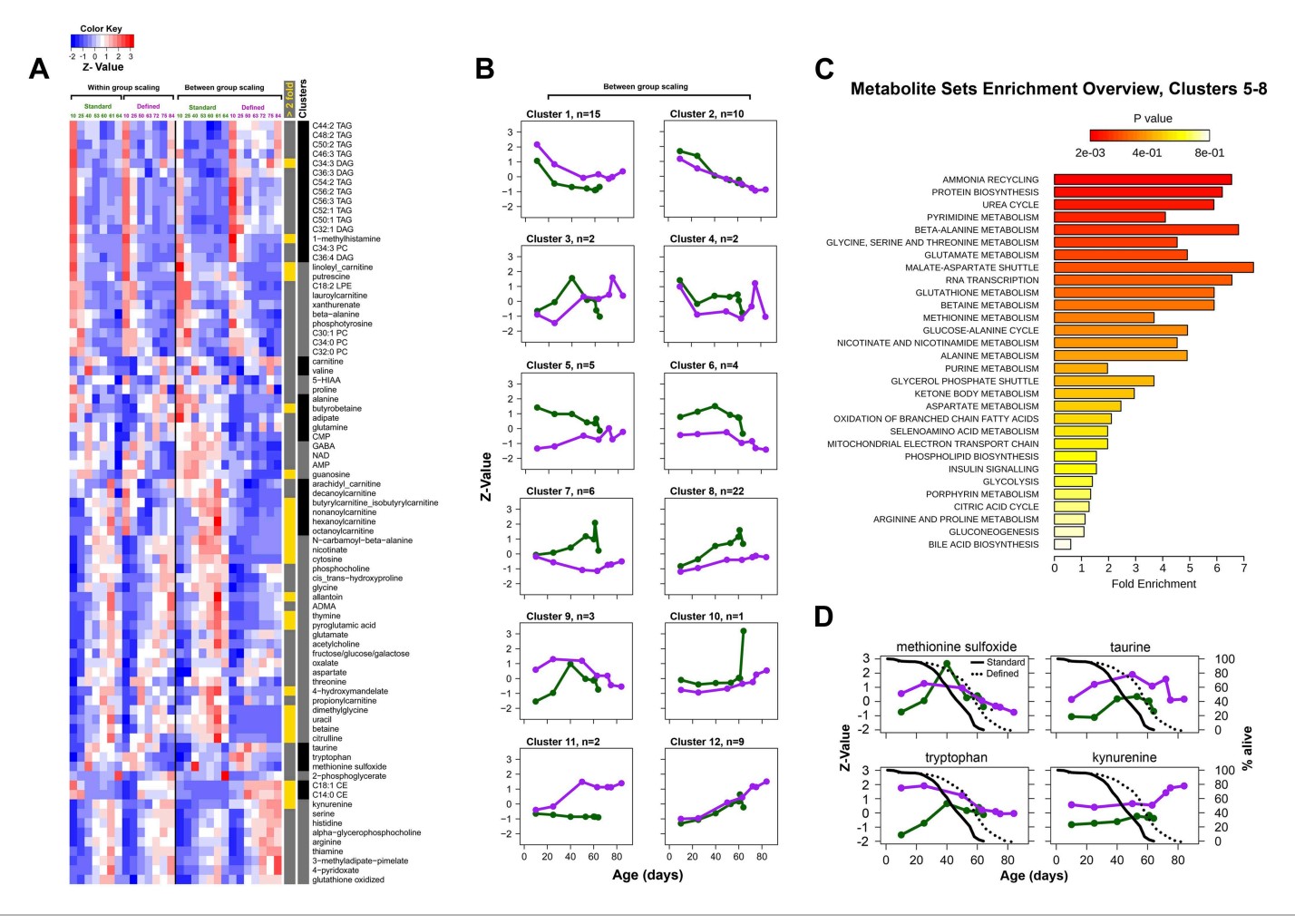

**Figure 7**. Metabolic signatures of aging in control and dietary restricted flies. Annotated (targeted) metabolites were derived from raw nontargeted data and represent only signals of established chemical identities. (**A** and **B**) Patterns of targeted metabolites. Clustering and graphing were done identically to the procedures described for *Figure 3C* legends. Side bars highlight lipid species and cluster boundaries that correspond to consequently arranged plots in (**B**). (**C**) Metabolite set enrichment analysis was performed by MetaboAnalyst 2.0 (*Xia and Wishart, 2011*). The panel overviews low expressing signals in long-living flies (Clusters 5–8). (**D**) Metabolites representing known damage and lifespan limiting factors overlayed with lifespan curves for standard (solid) and defined (dotted) diets. Taurine and kynurenine showed statistically significant inter-dietary changes across lifespan. Methionine sulfoxide differed significantly between 10 and 25 day groups (Student *t* test p<0.50). Tryptophan showed no significant inter-dietary differences at 60–63 days. Circles correspond to sampled age groups, whereby z-scored expression values are generated from averages of randomly measured replicates representing separately sampled cohorts in standard (green) or defined (purple) diets.

the role of damage, the contribution of changes in the levels of purposely used metabolites remains unclear. An age-related remodeling has been described for transcriptome, epigenome, and proteome, but how these changes relate to each other and what the role of metabolome remodeling is in the progression of aging is not well understood.

Our report offers insights into the nature of aging through a comprehensive assessment of changes in a large number of endogenous small molecules throughout *Drosophila* lifespan. Our initial observations showed that global metabolite profiling effectively recapitulated distinct stages of the aging process and revealed metabolites that strongly correlate with organismal survivorship. Specifically, analysis of nontargeted features showed a robust increase in the number of detected molecules as a function of age, followed by cessation of the increase in late life. This pattern corresponded to the transitions in the lifespan curves and was also in concert with the observed decelerated metabolome remodeling in late life, altogether supporting the idea that aging subsides at advanced ages. On the

other hand, the increase in the number of detected metabolites clearly differed from general age-associated changes, and its commonalities in the two dietary conditions exclude the role of exogenous compounds. The extended metabolome composition may include by-products that affect viability and damage forms that reach and pass the detection thresholds during organism's aging (*Phoenix and de Grey, 2007*), but they likely represent only the tip of the iceberg of cumulative damage. Further advances in instrumentation and sensitivity of metabolite profiling should expose additional metabolites that characterize cumulative damage.

We further followed age-related trajectories of 1066 individual metabolites common to short- and long-living flies and found that the trajectories of most metabolites correlated between the two dietary groups, yet varied in magnitude from initial to final concentration changes, suggesting that differences in accumulation of individual metabolites, in addition to the overall pattern, were responsible for lifespan differences. A substantial fraction of these signals showed biologically-relevant differences during aging and correlated with survivorship of flies. Metabolite trajectories were gradual and continuous starting from early adulthood, with very few signals showing mid-life reversals. These results are generally consistent with the observations from the studies on transcript and protein profiles across diverse aging populations (*Zou et al., 2000*; *Pletcher et al., 2002*; *Somel et al., 2010*). Indeed, our analysis of age-associated gene expression involving both control and dietary restricted long-lived individuals indicates that metabolites and transcripts exhibit nearly identical frequencies of diet-dependent to diet-independent effects. Thus, metabolites, similarly to genes, may exhibit diverse lifespan-regulatory roles. Similarly to targeted metabolite profiling in mammals (*Tomas-Loba et al., 2013*), the gradual transitions in thousands of nontargeted metabolites may be used to derive a signature that predicts survivorship across variable lifespans. Consistent with this idea, we observed a delay in the age-associated changes in transcripts and metabolites in the case of longer-lived flies. Thus, a change in lifespan (here, in response to a dietary regimen, but this should also apply to longevity-related genetic manipulations and evolutionary processes that affect species lifespan) involves both a change in the expressed transcripts and metabolites at any time throughout lifespan and a delay in age-related changes for those transcripts and metabolites that commonly change with age in both conditions (or genotypes).

A significant overlap between metabolites that show age-associated changes between the diets can also identify the biological, rather than the chronological, aging component. Together with the component that reflects diet-specific patterns of metabolite change, it should help define the aging process in model systems and beyond. For instance, comparison of inter-dietary age-associated trajectories using known metabolites reveals that a fraction of downregulated signals were associated with slower metabolism in long-living flies. However, we also found that the molecular species with higher levels in long-living flies represented damage forms. While the inter-dietary differences for such increases were below twofold and varied in significance across age, the data nonetheless implies that lifespan extension may be associated with damage tolerance via compensatory mechanisms. In fact, long-living flies, when compared to control, featured a large set of nontargeted metabolites that increase at higher levels. How many of these metabolites represent damage will need to be determined through structural elucidation of nontargeted signals.

Overall, from the current study the aging process emerges as a gradual, dynamic metabolome remodeling that involves changes in the levels (both increases and decreases) in numerous cellular metabolites, delays in these changes in late life and under conditions that lead to longer lifespan, and accumulation of damage, whose condition-dependent cumulative effects may impact survivorship.

## Materials and methods

### *Drosophila* lifespan extension and sampling

Progenies used in aging assays and metabolite profiling experiments were prepared by mating wild-type animals of Canton-S background, which were backcrossed for seven generations. Aging progeny was grown on rich media (flystocks.bio.indiana.edu) at regular density. Newly eclosing F1 adult males were collected for 3 days, mixed and distributed within vials containing respective food for sampling at predetermined ages. The recipes for preparation of the defined diet have been described (*Troen et al., 2007*). Between 30 and 50 flies were sampled at each age, rapidly frozen in liquid nitrogen and stored at −80°C. Once all samples were collected they were immediately processed for LC-MS profiling.

## Metabolite profiling

Three LC-MS methods were used to measure polar metabolites and lipids in whole fly homogenates. Conditions for the analysis were set using a panel of routinely analyzed 293 standards. Polar and lipid-associated species were extracted from 7 and 2 flies, respectively, in three separate replicates which were ran in randomized order. All data were acquired using an LC-MS system comprised of a Nexera X2 U-HPLC (Shimadzu, Marlborough, MA) and a Q Exactive hybrid quadrupole orbitrap mass spectrometer (Thermo Fisher Scientific; Waltham, MA). Hydrophilic interaction liquid chromatography (HILIC) analyses of water soluble metabolites in the positive ionization mode were carried out by extracting 10 µl homogenate using 90 µl of 74.9:24.9:0.2 vol/vol/vol acetonitrile/methanol/formic acid containing stable isotope-labeled internal standards (valine-d8, Isotec; and phenylalanine-d8, Cambridge Isotope Laboratories; Andover, MA). The samples were centrifuged (10 min, 9000×$g$, 4°C) and the supernatants were injected directly onto a 150 × 2 mm Atlantis HILIC column (Waters; Milford, MA). The column was eluted isocratically at a flow rate of 250 µl/min with 5% mobile phase A (10 mM ammonium formate and 0.1% formic acid in water) for 1 min followed by a linear gradient to 40% mobile phase B (acetonitrile with 0.1% formic acid) over 10 min. The electrospray ionization voltage was 3.5 kV and data were acquired using full scan analysis over m/z 70–800 at 70,000 resolution and a 3 Hz data acquisition rate. Negative ion mode analyses of polar metabolites were achieved using a HILIC method under basic conditions. Briefly, 30 µl homogenate was extracted using 120 µl of 80% methanol containing inosine-$^{15}$N4, thymine-d4, and glycocholate-d4 internal standards (Cambridge Isotope Laboratories; Andover, MA). The samples were centrifuged (10 min, 9000×$g$, 4°C) and the supernatants were injected directly onto a 150 × 2.0 mm Luna NH2 column (Phenomenex; Torrance, CA) that was eluted at a flow rate of 400 µl/min with initial conditions of 10% mobile phase A (20 mM ammonium acetate and 20 mM ammonium hydroxide in water) and 90% mobile phase B (10 mM ammonium hydroxide in 75:25 vol/vol acetonitrile/methanol) followed by a 10 min linear gradient to 100% mobile phase A. MS full scan data were acquired over m/z 70–800. The ionization source voltage is −3.0 kV and the source temperature is 325°C. Lipids were extracted from 10 µl of homogenate using 190 µl of isopropanol containing 1-dodecanoyl-2-tridecanoyl-sn-glycero-3-phosphocholine (Avanti Polar Lipids; Alabaster, AL). After centrifugation, supernatants were injected directly onto a 150 × 3.0 mm Prosphere HP C4 column (Grace, Columbia, MD). The column was eluted isocratically with 80% mobile phase A (95:5:0.1 vol/vol/vol 10 mM ammonium acetate/methanol/acetic acid) for 2 min followed by a linear gradient to 80% mobile-phase B (99.9:0.1 vol/vol methanol/acetic acid) over 1 min, a linear gradient to 100% mobile phase B over 12 min, then 10 min at 100% mobile-phase B. Full scan MS analyses (m/z 400–1000) were carried out in the positive ion mode using full scan analysis at 70,000 resolution and 3 Hz data acquisition rate. All raw data were processed using Progenesis CoMet software (version 2.0, NonLinear Dynamics) for feature alignment, signal detection, and signal integration. Signal peak areas were converted into numerical intensity values and normalized to internal standards added to each sample and to total signal at each time point. The raw peak intensity values are provided as Supplementary dataset (*Avanesov et al., 2014*).

## Statistical analysis

Age-associated signals were identified using repeated measures ANOVA, and significance was established at $p<0.0014$ (<0.2% of false positives) and False discovery rate (FDR) of 0.029. To calculate FDR, we first established relationships between error rate and its corresponding p value (*Storey and Tibshirani, 2003*) and used an adjusted p value at the interface that marked transition between linear and exponential rise in error rate. Then, FDR was calculated using compute.FDR function in R brainwaver package with the significance thresholds established above. Data analysis was also carried on in the R environment; information on packages used in this study can be found within respective figure legends. Metabolite enrichment analysis was performed in MetaboAnalyst 2.0 (*Xia and Wishart, 2011*). For gene expression analysis, we used a dataset that was normalized using RMA package of Bioconductor in R and kindly provided by Dr Scott Pletcher (*Pletcher et al., 2002*). Age-associated gene expression signatures were identified identically to the metabolites above and corrected for multiple testing ($p<0.0013$, FDR = 0.025) using aforementioned computations and significance threshold selections.

## Acknowledgements

We thank Dr Scott Pletcher for providing transcript expression data and lab members for comments and stimulating discussions. This work was supported by NIH AG038004 and AG021518 to VNG and GM109312 to SHY. Funding agency had no involvement in study design, data collection and interpretation.

# Additional information

## Funding

| Funder | Grant reference number | Author |
|---|---|---|
| National Institutes of Health | AG038004, AG021518 | Vadim N Gladyshev |
| National Institutes of Health | GM109312 | Sun Hee Yim |

The funders had no role in study design, data collection and interpretation, or the decision to submit the work for publication.

## Author contributions

ASA, KAP, CBC, Acquisition of data, Analysis and interpretation of data, Drafting or revising the article; SM, Analysis and interpretation of data, Drafting or revising the article; SHY, BCL, Analysis and interpretation of data, Drafting or revising the article, Contributed unpublished essential data or reagents; VNG, Conception and design, Analysis and interpretation of data, Drafting or revising the article

# Additional files

## Major dataset

The following dataset was generated:

| Author(s) | Year | Dataset title | Dataset ID and/or URL | Database, license, and accessibility information |
|---|---|---|---|---|
| Avanesov AS, Ma S, Pierce K, Yim SH, Lee BC, Clish CB, Gladyshev VN | 2014 | Data from: Age- and diet-associated metabolome remodeling characterizes the aging process driven by damage accumulation | http://dx.doi.org/ 10.5061/dryad.nn5qc | Publicly available at the Dryad digital repository. |

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
