## [Decision Letter]

Thank you for sending your work entitled “Age- and diet-associated metabolome remodeling characterizes the aging process driven by damage accumulation” for consideration at *eLife*. Your article has been evaluated by a Senior editor, a Reviewing editor, and 2 reviewers.

The Reviewing editor and the other reviewers discussed their comments before we reached this decision, and the Reviewing editor has assembled the following comments to help you prepare a revised submission.

One of the major concerns voiced by both reviewers and shared by the editor is that this work is strongly interpreted by the authors to support the aging hypothesis recently formulated by the senior author of the manuscript. While such a trend is understandable, it is not obvious that the presented data cannot be explained by other aging hypotheses. Furthermore, authors would be advised to try to provide more evidence that metabolome changes and survival changes are really as well correlated as claimed. For instance, to provide additional support for the aging hypothesis promoted by the authors, one may need to show that differences in survival curves between dietary controlled animals and common animals can be linked with differences in the shapes of metabolite repertoire curves. It is not clear however, whether data provided by the authors could unambiguously support such a claim. Thus, in the absence of additional evidence, a much more balanced data interpretation considering other aging hypotheses would be advised.

Other issues to address:

*Reviewer 1*:

In this work, the authors performed metabolic profiling in aging flies and also in context of caloric restriction. To my knowledge, this is the first metabolic profiling of aging in flies, and one of the first in context of aging, and therefore will be of interest to researchers in the field. The work is largely sound, though some aspects, including the methodology, require further clarification, and at times the authors' interpretations of the results and how it fits our current knowledge of aging were unclear to me, as detailed below.

One concern I have is that there are aspects of the methods that I think require further clarification. One aspect that wasn't clear to me is whether the authors used females and males? Figure 1 only refers to males but if the authors only used males then they should explicitly mention this in the manuscript. Also, the authors write that: “As metabolites were extracted from the same number of flies at each time point, the decreased total signal may represent lower biomass in older flies”; but then shouldn't the results be normalized by biomass or by the total signal? I'm not an expert in metabolic profiling, and thus could be mistaken, but this seems reasonable to me. Also, the authors write: “Using repeated measures ANOVA and stringent statistical cut-off that estimated <0.2% of false positives (Materials and Methods)”; but I couldn't find where this is described in the Methods. A reference is made to an FDR of 0.029 which ∼3%. It also wasn't clear to me how FDR was estimated and significance established.

I think the authors can do a better job of placing their findings in context of current knowledge. There are several statements, in particular in the Introduction, that are not supported by a suitable reference. For example, the authors mention how “it is also known that lifespan can actually be increased under mild stress”, which is called hormesis by various authors with important contributions in this context (e.g., Rattan). Also: “it is not known whether cumulative damage increases linearly or exponentially throughout lifespan”; yet various authors have quantified different types of damage (DNA mutations in aging tissues in mice by Vijg and co-workers, more recent quantifications of DNA mutations in human samples with NGS, etc).

I thought some statements and conclusions by the authors were unclear and/or not entirely supported by the data. For example: “This finding suggested that the observed metabolite pattern characterized cumulative damage.” But have the authors considered that many changes with age are adaptive responses trying to fend off aging, as suggested by gene expression studies (e.g., see de Magalhaes et al. Bioinformatics 25:875-881)? I may be missing something but the interpretation of the results in context of cumulative damage wasn't clear to me. In the Discussion the authors write: “Aging was suggested to result from accumulation of molecular damage...” and a few sentences later: “Alternatively, aging may be due to cumulative damage”; why should this be an alternative? “Accumulation of molecular damage” and “cumulative damage” mean the same to me.

Also: “Thus, the observed gradual, dynamic metabolome remodeling throughout lifespan, including both increases and decreases in metabolite signals, implied a role of predetermined, genetically-defined aging.” It wasn't clear to me why this should be the case. Besides, even assuming this is the case, such genetic regulatory basis in aging has been shown by other studies, which should be mentioned (for a recent review see de Magalhaes, FASEB Journal 26:4821-4826; also Somel 2010 which is cited later in the manuscript but should also be mentioned in this context).

“These metabolite analyses provide critical insights into the nature of the aging process.” While this type of work is novel, I think this is an overstatement. Many of the results obtained by the authors are in line with overall patterns and signatures of aging and CR, for example at the level of lipid metabolism and stress response genes (e.g., Plank et al. Molecular BioSystems 8:1339-1349). The authors should either be more clear about the insights from by their results or tone down their interpretation.

I would also suggest that the authors check the overlap between their gene expression results and the GenDR database of CR-related genes since these are genes demonstrated directly to affect CR-mediated longevity and may allow the authors to identify candidate regulators of CR longevity effects: http://genomics.senescence.info/diet/

Lastly, the authors write: “only a small fraction of metabolites showed more than 2-fold changes between the two diets”; are these given in the supplementary material? In fact, is the full data available as supplementary material or in a suitable repository?

Overall, this is an original and interesting work, though some aspects of the manuscript may be improved and clarified. It is possible that some of my comments reflect misunderstandings of mine. If so then I would suggest that the author uses my misunderstandings as an indication that such points might be made clearer in the manuscript.

It my usual policy to reveal my identity to the authors: Joao Pedro de Magalhaes.

*Reviewer 2*:

Avanesov et al describe a metabolome analysis of fruit fly aging, using both a standard and lifespan extending restricted diet. They find that 15% of metabolites levels change during aging; that the timing and speed of metabolite level changes during aging largely parallels those of mRNA, and is slower under lifespan extending dietary conditions; that the number of detected metabolites increases with age; that metabolites in the lipid fraction decrease with age and that this decrease is unrelated to diet effect/longevity. Also, interestingly, the rate of metabolome changes is decelerated during aging, though such deceleration is not observed in gene expression (although the latter data is from another experiment). The authors further find higher levels of potentially damage-representing metabolites in long-lived DR flies, than standard diet fed flies, which they suggest could be explained by extra stress resistance of DR flies.

Major points:

The data and the methods appear sound. My main objection is regarding how some results were interpreted. For me, the claims that metabolome changes are directly linked to mortality, and that metabolome changes reflect cumulative damage, are both overinterpretations. Specifically:

1) “This transition in the metabolome diversity corresponded to the late-life transition in the survival curves, linking metabolome diversity with the aging process. Thus, while damage accumulation is continuous throughout adulthood, it decelerates at advanced stages of life.”

For me, this does not link “metabolome diversity with the aging process”, but only suggests a link. Second, how is it concluded that metabolome diversity reflects damage accumulation? This should be better explained, or the claim softened.

2) “The observed changes in the metabolome uncovered a link between the aging process and metabolite diversity. They also suggested that many metabolites reaching detection in the course of aging may represent low levels of by-products or other damage forms.”

Again, based on what?

3) “The observed pattern provided additional insights into damage accumulation. For example, the number of newly detected species did not rise exponentially or linearly during aging, instead reflecting deceleration of aging in late life.”

Again, this only assuming that the change in metabolite number reflects damage accumulation.

4) “This data is also congruent with the damage pattern (Figure 1) and confirms the links between activity-dependent damage accumulation and aging.”

5) “This pattern matched transitions in the mortality curves (...)”.

In order to claim this, the authors should first clearly define “transitions in the mortality curves”; i.e., the ages when transitions happen. They could then show that points when metabolite number changes or metabolite intensity level changes decelerate or accelerated correspond to these mortality transitions.

In summary, the general claim here is that because the change in metabolite number is correlated with mortality, the former should reflect molecular damage. This involves a lot of assumptions (which are not openly stated) and can only be a suggestion, not an observation.

Second, I am not sure if the mortality curves indeed are correlated with the change in detected metabolite number. In Figure 1, the increase in metabolite diversity appears before mortality increases, it slows down when mortality increases. So I am not sure whether it could be concluded that deceleration in metabolite diversity refletcs “deceleration of aging in late life”.

Perhaps the authors have data to support these claims, but then these should be more clearly explained.

---

## [Author Response]

*One of the major concerns voiced by both reviewers and shared by the editor is that this work is strongly interpreted by the authors to support the aging hypothesis recently formulated by the senior author of the manuscript. While such a trend is understandable, it is not obvious that the presented data cannot be explained by other aging hypotheses. Furthermore, authors would be advised to try to provide more evidence that metabolome changes and survival changes are really as well correlated as claimed. For instance, to provide additional support for the aging hypothesis promoted by the authors, one may need to show that differences in survival curves between dietary controlled animals and common animals can be linked with differences in the shapes of metabolite repertoire curves. It is not clear however, whether data provided by the authors could unambiguously support such a claim. Thus, in the absence of additional evidence, a much more balanced data interpretation considering other aging hypotheses would be advised*.

We revised the manuscript to provide a more balanced interpretation of the data and distance the manuscript from the recently formulated aging hypothesis. Instead, we elaborate on the widely accepted ideas of damage accumulation. The Introduction was also modified to reflect additional studies. We toned down conclusions based on the reviewers’ comments. We have carried out additional analyses and provide new figures and panels in support of the changes in metabolite diversity and low-abundance metabolites. We hope that the responses to specific comments below and changes in the text fully address the issues identified during the review process.

*Other issues to address*:

Reviewer 1:

*In this work, the authors performed metabolic profiling in aging flies and also in context of caloric restriction. To my knowledge, this is the first metabolic profiling of aging in flies, and one of the first in context of aging, and therefore will be of interest to researchers in the field. The work is largely sound, though some aspects, including the methodology, require further clarification, and at times the authors' interpretations of the results and how it fits our current knowledge of aging were unclear to me, as detailed below*.

*One concern I have is that there are aspects of the methods that I think require further clarification. One aspect that wasn't clear to me is whether the authors used females and males?*
Figure 1
*only refers to males but if the authors only used males then they should explicitly mention this in the manuscript*.

We used males. We now indicate this in three additional places (end of Introduction, beginning of Results and Methods).

*Also, the authors write that: “As metabolites were extracted from the same number of flies at each time point, the decreased total signal may represent lower biomass in older flies”; but then shouldn't the results be normalized by biomass or by the total signal? I'm not an expert in metabolic profiling, and thus could be mistaken, but this seems reasonable to me*.

In the Methods section, we added information on normalization: “Signal peak areas were converted into numerical intensity values and normalized to internal standards added to each sample and to total signal at each time point.” All of our analyses besides Figure 1 used such normalized data. Our intention for presenting the original data in Figure 1 was to point out any obvious trends in total signal as well as its relation to the increase in metabolite diversity. We believe this information is most useful to the readers. The main point is that the increased metabolite diversity is not due to increased signal.

*Also, the authors write: “Using repeated measures ANOVA and stringent statistical cut-off that estimated < 0.2% of false positives (Materials and Methods)”; but I couldn't find where this is described in the Methods. A reference is made to an FDR of 0.029 which ∼3%. It also wasn't clear to me how FDR was estimated and significance established*.

We added the description of tools used in calculating FDR in Methods: “To calculate FDR, we first established relationships between error rate and its corresponding p value (51) and used an adjusted p value at the interface that marked transition between linear and exponential rise in error rate. Then, FDR was calculated using compute.FDR function in the R brainwaver package with the significance thresholds established above.”

*I think the authors can do a better job of placing their findings in context of current knowledge. There are several statements, in particular in the Introduction, that are not supported by a suitable reference. For example, the authors mention how “it is also known that lifespan can actually be increased under mild stress”, which is called hormesis by various authors with important contributions in this context (e.g., Rattan)*.

This section was significantly revised. We added references to Rattan (46; 47) and others (1; 15) on damage conditioning and also on the limited number of damage markers used to assess the damage pattern as function of age. We also emphasized that earlier studies mostly left out late-life time points and were unable to address the remodeling pattern thoroughly.

*Also: “it is not known whether cumulative damage increases linearly or exponentially throughout lifespan”; yet various authors have quantified different types of damage (DNA mutations in aging tissues in mice by Vijg and co-workers, more recent quantifications of DNA mutations in human samples with NGS, etc)*.

The reviewer is correct about a large body of existing data characterizing damage and damage response during aging. We mentioned these studies in the Introduction in our revision. However, to our knowledge these studies never analyzed late life changes and this can lead to elusive assumptions about life-long changes in damage rates and responses. In addition, selected damage markers (e.g. DNA mutation rate, lipofuscin accumulation, ROS levels, etc.) are insufficient for assessing cumulative damage and their resolution is too low at the moment to conclude on the pattern of changes (beyond the fact that damage increases with age). In our study, we attempt to address these deficiencies by using non-targeted metabolite profiling and including multiple late time points to decipher the patterns of metabolite damage and their relationship to metabolism. We revised the text to better explain this point.

*I thought some statements and conclusions by the authors were unclear and/or not entirely supported by the data. For example: “This finding suggested that the observed metabolite pattern characterized cumulative damage.” But have the authors considered that many changes with age are adaptive responses trying to fend off aging, as suggested by gene expression studies (e.g., see de Magalhaes et al. Bioinformatics 25:875-881)*?

This is an important point as we may be the first to employ metabolite diversity in making inferences about damage patterns. We are familiar with the nice meta-analysis study mentioned by the reviewer as well as with another one by Plank et al. and also with the datasets from Pletcher, Tower and Khaitovich groups and many others. Similarly, studies on aging DNA methylomes (e.g., Hannum et al. Mol Cell 2013) show bi-directional changes, for which additional evidence implicates some of them as adaptive. However, unlike bi-directional changes in the regulatory and/or responsive pathways the damage pattern is largely unidirectional and is always rising (e.g., oxidative damage, somatic DNA mutations, protein aggregates, etc. all rise with age). In our study, we show that metabolite diversity rises with age and the rises are largely represented by low abundance molecules. These considerations argue that the observed changes in metabolite diversity expose cumulative damage. In contrast, an analysis of non-targeted metabolites with steady diversity (ones detected in all ages) and known metabolites shows that they have bi-directional changes. We can only argue that this second type of change in metabolomes reflects biological activity (possibly adaptive responses too) and they should not be fully regarded as damage. We have made a distinction between the two by adding the following statement (Results): ”Among signals consistently detected in all aging samples (those showing no difference in diversity) there may be metabolites that are passengers or drivers of the aging process which would be similar to the genes with age-associated differential expression (30; 13; 44)” We also commented on a possibility of adaptive changes contributing to the observed changes in metabolites.

*I may be missing something but the interpretation of the results in context of cumulative damage wasn't clear to me. In the Discussion the authors write: “Aging was suggested to result from accumulation of molecular damage...” and a few sentences later: “Alternatively, aging may be due to cumulative damage”; why should this be an alternative? “Accumulation of molecular damage” and “cumulative damage” mean the same to me*.

To clarify, we replaced “alternatively” with “however”. “Alternatively” referred to the original sentence which discussed individual damage forms, contrasting them with cumulative damage. Cumulative damage combines individual damage forms that accumulate, but often cannot be defined as causal factors. Many components of cumulative damage would have a very minor role when taken in isolation. For example, five molecules of a by-product produced by an enzyme during lifetime of a cell would not be “seen” by evolution, would not be detected by analytical methods, and would not show measurable role in aging when taken in isolation, but they would contribute, with the millions of other five-molecule by-product forms, to cumulative damage. Of course, this is how we think about the aging process, but we made sure the data interpretation in the manuscript is not biased by these ideas, which have been removed during revision.

*Also: “Thus, the observed gradual, dynamic metabolome remodeling throughout lifespan, including both increases and decreases in metabolite signals, implied a role of predetermined, genetically-defined aging.” It wasn't clear to me why this should be the case. Besides, even assuming this is the case, such genetic regulatory basis in aging has been shown by other studies, which should be mentioned (for a recent review see de Magalhaes, FASEB Journal 26:4821-4826; also Somel 2010 which is cited later in the manuscript but should also be mentioned in this context)*.

In addition to Somel et al. (50) on continuous transcript changes in aging primate brains, the revised manuscript also has additional citations (including the review mentioned by the reviewer). We also slightly modified the text to better explain this point.

*“These metabolite analyses provide critical insights into the nature of the aging process.” While this type of work is novel, I think this is an overstatement. Many of the results obtained by the authors are in line with overall patterns and signatures of aging and CR, for example at the level of lipid metabolism and stress response genes (e.g., Plank et al. Molecular BioSystems 8:1339-1349). The authors should either be more clear about the insights from by their results or tone down their interpretation*.

We modified that last sentence of the Introduction and made changes throughout the text to limit the use of critical to more specific context such as in the “...activity-driven nature of the aging process”. By measuring metabolites as function of age between distinct dietary conditions and by analyzing late life points we provide support for the patterns of metabolite change and damage and show that the changes slow down and then stop at the very advanced time points. Our studies also open the door to using changes in metabolite diversity as readout for cumulative damage in age-related studies, and further improvement in sensitivity of this technology should increase the fraction of damage over functional metabolites. In our opinion, such global approach allows for a more comprehensive detection of damage compared to the analysis of selected damage types or selected makers. Lastly, by analyzing the effects of dietary restriction we show that lifespan increase may be associated with an increase in specific damage types.

*I would also suggest that the authors check the overlap between their gene expression results and the GenDR database of CR-related genes since these are genes demonstrated directly to affect CR-mediated longevity and may allow the authors to identify candidate regulators of CR longevity effects*: *http://genomics.senescence.info/diet/*

Thank you. We examined the overlap between significant age-associated gene expression in our analysis (Figure 5) and the twenty Drosophila lifespan regulatory genes available at DenDR. llp5 was not represented in the gene expression dataset we used. As highlighted below, there is a good agreement between the genes that change expression with age and the genes required for the effect of DR. This encourages identification of lifespan regulating genes based on changes in their age-associated expression. We have also extended this comparison to Drosophila genes whose homologs affect lifespan in other species. As tabulated below, there is a greater overlap with such dataset in our manuscript (in Tables 1 and 2 below, we removed 16 genes whose expression was not mapped on our arrays).Author response table 1.The table was downloaded from the GenAge database and shows genes known to affect *D. melanogaster* lifespan. We asked if gene expression differences as function of age were significant for these *D. melanogaster* genes. We find that there is a good correlation as indicated by significant age-associated p values (highlighted cells).gene_symbolgene_idage-assoc p val. (Control diet)age-assoc p val. (DR diet)entrez_idgene_nameRpd3FBgn00158050.4818105470.21601182638565Histone deacetylase Rpd3Or83bFBgn00373240.285181195**0.046763434**40650Odorant receptor 83bchicoFBgn0024248**0.003887938**0.20551386864880Insulin receptor substrate-1gigFBgn00051980.4747975710.18066355140201gigasSir2FBgn00242910.321025147**0.003652163**34708Protein Sir2IndyFBgn00368160.113197528**0.006970164**40049I'm not dead yetfoxoFBgn00381970.281422096**0.046040677**41709Forkhead box, sub-group OThorFBgn0261560**9.13398E-09****3.2467E-07**33569p53FBgn00390440.2020609440.2776168462768677CG33336 gene product from transcript CG33336-RBl(3)neo18FBgn0011455**0.028431267****0.006074331**46260lethal (3) neo18CG11015FBgn00390440.2020609440.27761684633918Ilp2FBgn00360460.7609271940.09398943939150Insulin-like peptide 2Ilp3FBgn00440500.9585270520.69113422439151Insulin-like peptide 3Ilp5FBgn0044048not in the datasetnot in the dataset2768992Insulin-like peptide 5CbsFBgn0031148**0.000278535****0.000174874**33081Cystathionine beta-synthaseCG5389FBgn00365680.3188683090.17706720439761CG4389FBgn0028479**0.001757849****7.90528E-05**34276CG4389 gene product from transcript CG4389-RACG7834FBgn00396970.2193464440.11176124543515CG7834 gene product from transcript CG7834-RArprFBgn00117060.4284758520.08134098440015reaperAkhFBgn0004552**2.04158E-06**0.08810508138495Adipokinetic hormone

Author response table 2.The table was downloaded from the GenAge database and shows *D. melanogaster* genes whose homologs affect *C. elegans* and *S. cerevisiae* lifespans. We asked if gene expression differences as function of age were significant for these genes. We find that there is a good correlation as indicated by significant age-associated p values (highlighted cells).entrez_idgene_idmodel organism frommodel organism gene symbolmodel organism gene entrez_idage-assoc p val. (Control diet)age-assoc p val. (DR diet)37068FBgn0001222*Caenorhabditis elegans*hsf-11730780.208232323**0.014228222**32780FBgn0003380*Caenorhabditis elegans*shk-1174536**0.004195376**0.05899931233379FBgn0003557*Caenorhabditis elegans*wwp-11716470.1422173520.13063464942549FBgn0013984*Caenorhabditis elegans*daf-21754100.349618864**0.011412462**42446FBgn0015279*Caenorhabditis elegans*age-11747620.664739688**1.19E-05**43856FBgn0015624*Caenorhabditis elegans*cbp-11763800.335702205**0.000305516**33025FBgn0015789*Caenorhabditis elegans*rab-10266836**2.50E-05****0.002107404**37546FBgn0020307*Caenorhabditis elegans*dve-11803980.951050741**0.022711332**47396FBgn0021796*Caenorhabditis elegans*let-363172167**0.011455499****0.030110581**43904FBgn0023169*Caenorhabditis elegans*aak-21817270.4435647810.23011177837035FBgn0026316*Caenorhabditis elegans*ubc-181759850.3876128790.45876604444007FBgn0029502*Caenorhabditis elegans*clk-11757290.604169327**4.28E-05**31443FBgn0029752*Caenorhabditis elegans*trx-11818630.4343623980.3027816132864FBgn0030954*Caenorhabditis elegans*ckr-11887740.8398284860.06713543732864FBgn0030954*Caenorhabditis elegans*ckr-21753410.8398284860.06713543737141FBgn0034366*Caenorhabditis elegans*atg-71780050.5218779610.16231649942358FBgn0038736*Caenorhabditis elegans*ire-1174305**0.004739753****2.99E-05**117332FBgn0041191*Caenorhabditis elegans*rheb-1176327**2.57E-10****0.002436925**42162FBgn0003499*Saccharomyces cerevisiae*MSN48538030.3021405880.09295291141311FBgn0011768*Saccharomyces cerevisiae*SFA18513860.161108282**0.021733851**33418FBgn0014010*Saccharomyces cerevisiae*VPS218542560.364851406**0.001462461**42036FBgn0015230*Saccharomyces cerevisiae*HXT178558090.8788177360.47048010942841FBgn0015795*Saccharomyces cerevisiae*YPT78550120.140249034**0.00372342**38565FBgn0015805*Saccharomyces cerevisiae*RPD38553860.4818105470.21601182638654FBgn0015806*Saccharomyces cerevisiae*SCH98566120.6979579820.72579641247396FBgn0021796*Saccharomyces cerevisiae*TOR1853529**0.011455499****0.030110581**47611FBgn0022160*Saccharomyces cerevisiae*GUT2854651**0.023200204****0.021014678**45706FBgn0023541*Saccharomyces cerevisiae*ERG5855029**0.005058412****0.003531191**44098FBgn0024194*Saccharomyces cerevisiae*GUP1852796**0.000164407****0.004031428**31410FBgn0025679*Saccharomyces cerevisiae*MSN28550530.3239272210.69418306632768FBgn0030876*Saccharomyces cerevisiae*SRX1853776**3.16E-05**0.27979969533626FBgn0031589*Saccharomyces cerevisiae*NPT18543840.0686251750.48513759733837FBgn0031759*Saccharomyces cerevisiae*GIS18516700.6559293530.16358505934021FBgn0031912*Saccharomyces cerevisiae*LAT18556530.057376425**5.00E-07**37581FBgn0034744*Saccharomyces cerevisiae*VPS20855101**0.032323885****6.08E-05**38612FBgn0035600*Saccharomyces cerevisiae*CYT18542310.082518654**1.92E-08**38735FBgn0035704*Saccharomyces cerevisiae*VPS88512610.4096958120.1046757241071FBgn0037647*Saccharomyces cerevisiae*GTR18549180.440417904**0.031441517**41210FBgn0037761*Saccharomyces cerevisiae*SUR48510870.121178495**0.047327998**42185FBgn0038587*Saccharomyces cerevisiae*MDH18537770.201852989**0.005025531**53578FBgn0040309*Saccharomyces cerevisiae*TSA18549800.831741497**0.000946166**53581FBgn0040319*Saccharomyces cerevisiae*GSH18533440.3565150360.17451735438753FBgn0041194*Saccharomyces cerevisiae*ADE4855346**8.21E-06****2.59E-07**43191FBgn0042710*Saccharomyces cerevisiae*HXK28526390.367984828**0.002632023**42348FBgn0051216*Saccharomyces cerevisiae*PNC1852846**0.052873779**0.09505664332097FBgn0052666*Saccharomyces cerevisiae*PKH28540530.2045558820.40882749442850#N/A*Caenorhabditis elegans*bec-1177345not in the datasetnot in the dataset40633#N/A*Caenorhabditis elegans*egl-9179461not in the datasetnot in the dataset41612#N/A*Caenorhabditis elegans*hif-1180359not in the datasetnot in the dataset48552#N/A*Caenorhabditis elegans*sams-1181370not in the datasetnot in the dataset41675#N/A*Caenorhabditis elegans*smk-1179243not in the datasetnot in the dataset39454#N/A*Caenorhabditis elegans*unc-51180311not in the datasetnot in the dataset37733#N/A*Caenorhabditis elegans*vps-34172280not in the datasetnot in the dataset31618#N/A*Saccharomyces cerevisiae*CDC25851019not in the datasetnot in the dataset44297#N/A*Saccharomyces cerevisiae*DAP2856423not in the datasetnot in the dataset35494#N/A*Saccharomyces cerevisiae*FET3855080not in the datasetnot in the dataset3771854#N/A*Saccharomyces cerevisiae*HHF1852294not in the datasetnot in the dataset42414#N/A*Saccharomyces cerevisiae*HST2856092not in the datasetnot in the dataset326219#N/A*Saccharomyces cerevisiae*LCB4854342not in the datasetnot in the dataset318252#N/A*Saccharomyces cerevisiae*NFU1853826not in the datasetnot in the dataset35988#N/A*Saccharomyces cerevisiae*RPL31A851484not in the datasetnot in the dataset42850#N/A*Saccharomyces cerevisiae*VPS30855983not in the datasetnot in the dataset

*Lastly, the authors write: “only a small fraction of metabolites showed more than 2-fold changes between the two diets”; are these given in the supplementary material? In fact, is the full data available as supplementary material or in a suitable repository*?

We included a figure (Figure 3) with two separate heatmaps to show age-associated trends for metabolites with greater or less than 2-fold changes between the two diets. In that panel, the number of metabolites contributing to each heatmap is provided on the left. We also provide the entire dataset in Supplementary Materials. The data is presented in a user-friendly format featuring peak intensities for each metabolite across all time points for both diets. This file contains directions for normalization (including loaded controls), aside from description in Methods. The full dataset also features retention times, m/z values and identity of structurally resolved metabolites for further investigation.

*Overall, this is an original and interesting work, though some aspects of the manuscript may be improved and clarified. It is possible that some of my comments reflect misunderstandings of mine. If so then I would suggest that the author uses my misunderstandings as an indication that such points might be made clearer in the manuscript*.

*It my usual policy to reveal my identity to the authors: Joao Pedro de Magalhaes*.

We sincerely appreciate these insightful comments.

Reviewer 2:

*Avanesov et al. describe a metabolome analysis of fruit fly aging, using both a standard and lifespan extending restricted diet. They find that 15 % of metabolites levels change during aging; that the timing and speed of metabolite level changes during aging largely parallels those of mRNA, and is slower under lifespan extending dietary conditions; that the number of detected metabolites increases with age; that metabolites in the lipid fraction decrease with age and that this decrease is unrelated to diet effect/longevity. Also, interestingly, the rate of metabolome changes is decelerated during aging, though such deceleration is not observed in gene expression (although the latter data is from another experiment). The authors further find higher levels of potentially damage-representing metabolites in long-lived DR flies, than standard diet fed flies, which they suggest could be explained by extra stress resistance of DR flies*.

*Major points*:

*The data and the methods appear sound. My main objection is regarding how some results were interpreted. For me, the claims that metabolome changes are directly linked to mortality, and that metabolome changes reflect cumulative damage, are both overinterpretations. Specifically*:

*1) “This transition in the metabolome diversity corresponded to the late-life transition in the survival curves, linking metabolome diversity with the aging process. Thus, while damage accumulation is continuous throughout adulthood, it decelerates at advanced stages of life.*”

*For me, this does not link “metabolome diversity with the aging process”, but only suggests a link*.

We have modified the original expression to “suggest”, referring to the relationship between changes in metabolite diversity and aging. We clarify on why diversity represents damage below.

*Second, how is it concluded that metabolome diversity reflects damage accumulation? This should be better explained, or the claim softened*.

We offer additional explanation in the revised version and add new analyses (Figure 1 and Figure 1—figure supplement 1) which strengthen our interpretations. A similar point was made by Reviewer 1 (please see responses to Comment 3).

*2) “The observed changes in the metabolome uncovered a link between the aging process and metabolite diversity. They also suggested that many metabolites reaching detection in the course of aging may represent low levels of by-products or other damage forms.*”

*Again, based on what*?

As can be seen in the revised version (Figure 1) metabolites that registered at zero at least in one sample, when detected in remaining samples have peak intensities at few orders of magnitude lower compared to 205 known metabolites that were used as comparison markers. Thus, we think that these untargeted species (increasing in diversity) may also represent by-products of metabolism, for example, via enzymatic promiscuity.

*3) “The observed pattern provided additional insights into damage accumulation. For example, the number of newly detected species did not rise exponentially or linearly during aging, instead reflecting deceleration of aging in late life.*”

*Again, this only assuming that the change in metabolite number reflects damage accumulation*.

Our additional data provide support for the notion that the rise in the number of detected molecules represents damage, or at least the majority of it. These molecules are not random and present in diverse dietary regimens (yeast complete and DR-like defined medium). Unlike systemic changes that are bi-directional (increases and decreases) we observe a very negligible decrease in metabolite diversity and instead see a strong and prominent increase (Figure 1). This would be analogous to the age-associated rise in DNA mutations, except in our case we cover a much larger repertoire of molecular targets. This allows us to monitor the rate of cumulative damage, and this is more accurate that using limited set of damage markers reported in previous studies.

*4) “This data is also congruent with the damage pattern (*Figure 1*) and confirms the links between activity-dependent damage accumulation and aging.*“

In the revision, we have revised this statement by replacing “confirms” with “suggests”.

*5) “This pattern matched transitions in the mortality curves (...)”*.

*In order to claim this, the authors should first clearly define “transitions in the mortality curves”; i.e., the ages when transitions happen. They could then show that points when metabolite number changes or metabolite intensity level changes decelerate or accelerated correspond to these mortality transitions*.

We removed matched and replaced it with corresponds. We also attempted to illustrate transitions more clearly using a newly added Figure 1—figure supplement 1 serving as a support to the main Figure 1. This additional figure reveals better correlations and transitions after we superimpose changes in metabolite diversity and changes in mortality (the number of dead flies). .

*In summary, the general claim here is that because the change in metabolite number is correlated with mortality, the former should reflect molecular damage. This involves a lot of assumptions (which are not openly stated) and can only be a suggestion, not an observation*.

Our claim that metabolite diversity represents damage is better supported by the actual data on diversity which we explained in several places below and provided additional evidence for its support in Figure 1. Correlations with mortality were supportive since the rate of damage (as represented by changes in metabolite diversity) declined along with mortality (Figure 1—figure supplement 1).

*Second, I am not sure if the mortality curves indeed are correlated with the change in detected metabolite number. In*
Figure 1*, the increase in metabolite diversity appears before mortality increases, it slows down when mortality increases. So I am not sure whether it could be concluded that deceleration in metabolite diversity refletcs “deceleration of aging in late life”*.

Figure 1—figure supplement 1 in the revised manuscript better reveals a correlation between transition points of mortality and metabolome diversity curves. We suggest that deceleration of aging is reflected in reduced metabolic activity (bottom panel of Figure 5), which leads to reduced metabolite diversity.